# Prevalence and Characterization of Extended-Spectrum β-Lactamase- and Carbapenemase-Producing Enterobacterales from Tunisian Seafood

**DOI:** 10.3390/microorganisms10071364

**Published:** 2022-07-06

**Authors:** Mehdi Sola, Yosra Mani, Estelle Saras, Antoine Drapeau, Raoudha Grami, Mahjoub Aouni, Jean-Yves Madec, Marisa Haenni, Wejdene Mansour

**Affiliations:** 1Laboratoire de Recherche Biophysique Métabolique et Pharmacologie Appliquée (LR12ES02), Faculté de Médecine Ibn Al Jazzar Sousse, Université de Sousse, Sousse 4002, Tunisia; solamehdi90@gmail.com (M.S.); yosramani1989@gmail.com (Y.M.); raoudhagr2016@gmail.com (R.G.); wejdene.mansour@gmail.com (W.M.); 2Unité Antibiorésistance et Virulence Bactériennes, ANSES Laboratoire de Lyon, Université de Lyon, 69007 Lyon, France; estelle.saras@anses.fr (E.S.); antoine.drapeau@anses.fr (A.D.); jean-yves.madec@anses.fr (J.-Y.M.); 3Laboratoire des Maladies Transmissibles et Substances Biologiquement Actives, Faculté de Pharmacie de Monastir, Université de Monastir, Monastir 5000, Tunisia; aouni_mahjoub2005@yahoo.com

**Keywords:** fish, clam, *bla*
_CTX-15_, *bla*
_NDM-1_, *bla*
_OXA-48_, IncF

## Abstract

Aquaculture is a rapidly expanding sector in which it is important to monitor the occurrence of multi-drug resistant (MDR) bacteria. The presence of extended-spectrum β-lactamase (ESBL-) or carbapenemase-producing Enterobacterales is a commonly used indicator of the resistance burden in a given sector. In this study, 641 pieces of farmed fish (sea bream and sea bass), as well as 1075 Mediterranean clams, were analyzed. All ESBL- and carbapenemase-producing Enterobacterales collected were whole-genome sequenced. The proportion of ESBL-producing Enterobacterales was 1.4% in fish and 1.6% in clams, carried by *Escherichia coli* (*n* = 23) and *Klebsiella pneumoniae* (*n* = 4). The ESBL phenotype was exclusively due to the presence of *bla*_CTX-M_ genes, the most frequent one being *bla*_CTX-M-15_. The *bla*_CTX-M-1_ gene was also identified in six *E. coli*, among which four were carried by IncI1/pST3 plasmids, possibly betraying an animal origin. Carbapenemases were absent in fish but identified in two *K. pneumoniae* isolates from clams (*bla*_NDM-1_ and *bla*_OXA-48_). Several sequence types (STs) identified were associated with human MDR clones such as *E. coli* ST131 and ST617, or *K. pneumoniae* ST307 and ST147. Our results might indicate that bacteria from hospital or farm effluents can reach the open sea and contaminate seafood and fish that are living or raised nearby. Therefore, monitoring the quality of water discharged to the sea and the presence of MDR bacteria in seafood is mandatory to ensure the quality of fishery products.

## 1. Introduction

Antimicrobial resistance (AMR) has emerged as one of the leading public health threats of the 21st century. In order to define efficient levers of action to lower this burden, it is of utmost importance to assess all sources of multi-drug resistant (MDR) bacteria, particularly in contexts where surveillance data are rare. The presence of *E. coli* producing extended-spectrum beta-lactamases (ESBLs), and to a lesser extent carbapenemases, has recurrently been reported in food-producing animals and food products [1,2]. As an example, the poultry sector has been under particular scrutiny due to elevated proportions of ESBL-producing Enterobacterales both in chicken and chicken meat [3,4,5]. On the contrary, the presence of MDR Enterobacterales in seafood products has been much less described, although studies in this field show the interest of an increased surveillance [2,6,7,8,9].

Aquaculture is a rapidly growing field of food production since the demand for fish is increasing worldwide, including in Tunisia where the sectors of fisheries and aquaculture play an important socio-economic role and contribute to about 9% of the value of agriculture [10]. Tunisia occupies a central place in the Mediterranean area and has more than 1300 km of coastline. Both marine and inland species are currently being farmed and the main aquaculture production zone is in the governorate of Sousse. Most of the shellfish production (mussels and oysters) comes from northern Tunisia (governorate of Bizerte) while Mediterranean clams are mostly cultured in the south-east governorate of Gabes. Tunisia’s aquaculture products are sold on the local and international markets, i.e., in Europe and America [10]. Consequently, MDR bacteria no longer have borders and, in addition to the local impact, can be disseminated abroad. Such a dissemination has been demonstrated for ESBL-producing *E. coli* that contaminated Brazilian chicken meat and was imported to Swedish, English, and Japanese markets [11,12,13].

In this context, the objectives of this study were to estimate the proportion of ESBL-and carbapenemase-producing Enterobacterales in seafood in Tunisia and to molecularly characterize the collected isolates.

## 2. Materials and Methods

### 2.1. Seafood Sampling and Isolation of Antimicrobial Resistant Bacteria

Two types of seafood products were sampled in unrelated markets in four different regions in Tunisia.

*Fish sampling.* A total of 641 pieces of farmed fish were purchased in three different markets (R1-R3) in central Tunisia (Sousse, Mahdia, and Monastir) between March 2014 and June 2015. R1 and R3 are open sea farms, while R2 is a farm in closed tanks. The sampling was composed of sea bream (*Sparus aurata*, 485 pieces) and sea bass (*Dicentrarchus labrax*, 156 pieces). Once purchased, all samples were placed on ice and immediately transported to the laboratory. The intestine of each fish was placed in tubes containing 10 mL of peptone salt broth, homogenized, and incubated for 24 to 48 h at 37 °C.

*Clam sampling*. Between March and April 2016, 1075 Mediterranean clams (*Ruditapes decussatus*) were purchased in markets in Gabès, in the southeast of Tunisia. Clam samples were aseptically transported at 4 °C to the laboratory and immediately processed. After removal of shell debris and algae, bivalves were dried, disinfected (70% ethanol), opened using a sterilized scalpel, and incubated in tubes containing peptone salt broth for 24 h at 37 °C. Each tube contained a pool of five pieces (215 pools in total).

After incubation, overnight cultures were inoculated on selective MacConkey agar plates supplemented with either imipenem or cefotaxime (final concentration of 2 mg/L), and one colony per morphology and per plate was picked up. Identification was performed using API20E test strips (bioMérieux, Marcy-l’Étoile, France) and confirmed by MALDI-TOF MS.

### 2.2. Antimicrobial Susceptibility Testing and ESBL Screening

Antimicrobial susceptibility was determined by the disk diffusion method on Mueller–Hinton agar plates. Sixteen β-lactam and 14 non-β-lactam antibiotics (Appendix A) were tested (Mast Diagnostics, Amiens, France) and the results were interpreted according to the guidelines of the Antibiogram Committee of the French Society for Microbiology (CA-SFM) [14]. *E. coli* ATCC 25922 was used as a quality control strain. ESBL-producing Enterobacterales were detected using the Double Disc Synergy Test (DDST) [15]. Carbapenem resistance was detected using an ertapenem 10 μg disk and was respectively confirmed using the ROSCO KPC/MBL and OXA-48 Confirm Kit (ROSCO Diagnostica, Taastrup, Denmark).

### 2.3. Phylogeny and Clonality

The major *E. coli* phylogenetic groups (A, B1, B2, or D) were identified according to Doumith et al. [16]. PFGE was performed using the restriction enzyme *Xba*I. Electrophoresis was conducted in a CHEF Mapper XP system using 6 V/cm at 14 °C for 24 h, with pulse times ramping from 10 to 60 s using an angle of 120 °C. PFGE results were interpreted according to international recommendations.

### 2.4. Genome Extraction, Sequencing and Assembly

DNA was extracted using a NucleoSpin Microbial DNA extraction kit (Macherey-Nagel, Hoerdt, France), according to the manufacturer’s instructions. Library preparation was performed using Nextera XT technology and sequencing was performed on a NovaSeq 6000 instrument (Illumina, San Diego, CA, USA). After sequencing, reads were quality trimmed and de novo assembled using Shovill v1.0.4 and the quality of assemblies was assessed using QUAST v5.0.2. Quality control statistics of all sequenced isolates are provided in Appendix A. All genomic sequences were deposited in DDBJ/EMBL/GenBank under the BioProject accession number PRJNA841857. Sequence types (STs) and resistance genes were determined using the CGE online tools (http://www.genomicepidemiology.org/, last accessed on 21 June 2022).

### 2.5. Molecular Characterization of Plasmids

The replicon content and plasmid formula were identified from the WGS data using PlasmidFinder 2.0.1 and pMLST 2.0 (http://www.genomicepidemiology.org/, last accessed on 21 June 2022). When in silico data showed no co-occurrence of resistance and replicon markers on the same contig, plasmids carrying the ESBL and carbapenemase genes were detected using Southern blot (using adequate probes tagging the genes and plasmids of interest) on PFGE-S1 gels (6 V/cm for 20 h with an angle of 120° at 14 °C with pulse times ranging from 1 to 30 s) [17]. When plasmidic location could not be evidenced, the chromosomal location was looked for by PFGE on *I-Ceu*1-digested DNA, followed by Southern blot hybridization using a 16S rDNA probe and probes corresponding to the ESBL and carbapenemase genes. Detection was performed using a DIG DNA Labeling and Detection Kit (Roche Diagnostics, Meylan, France) according to the manufacturer’s instructions.

## 3. Results

### 3.1. Proportion of ESBL- and Carbapenemase-Producing Isolates

Only *E. coli* and *K. pneumoniae* were retrieved from selective plates.

From farmed fish, nine ESBL-producing strains (9/641, 1.4%) were isolated, which were identified as *E. coli* (*n* = 6) and *K. pneumoniae* (*n* = 3) (Table 1). Only one isolate (#40598) was collected from sea bass (1/156, 0.6%), while the eight remaining isolates were from sea bream (8/485, 1.6%). Two *K. pneumoniae* belonged to ST983 and the third one belonged to ST13. Among *E. coli* (*n* = 6), three different STs were identified: ST617, ST10, and ST8149. ST10 (*n* = 2; isolated from the same retail market at different time points) and ST617 (*n* = 3; originating from three different markets) were clonal as determined by PFGE. All *E. coli* belonged to phylogroup A, considered normally non-pathogenic [18].

Among the 215 pools of 5 clams analyzed, 18 ESBL-producing isolates were identified, including 14 *E. coli* and 4 *K. pneumoniae* (Table 2). Each isolate was identified from one individual pool of clams so that the total proportion of ESBL-producing Enterobacterales can be approximated at around 1.6% (18/1075). *K. pneumoniae* isolates belonged to the ST17 (*n* = 1), ST307 (*n* = 1) and ST147 (*n* = 2). The ST17 and one ST147 isolate were also resistant to carbapenems (2/1075, 0.2%). *E. coli* isolates belonged to nine different STs, among which ST617, ST38, ST131, and ST2253 were found at least at two time points.

### 3.2. Characterisation of ESBL and Carbapenemase Genes

In fish, the ESBL phenotype was due to the presence of the *bla*_CTX-M-15_ gene in all nine isolates (Table 1). No carbapenemase gene was identified, as expected by the susceptible phenotype observed for ertapenem. The *bla*_CTX-M-15_ gene was carried on the chromosome in the two clonal ST10 isolates, while it was located on plasmids in the seven other isolates, belonging either to the IncF type or being untypable.

In clams, the ESBL phenotype was also uniquely due to CTX-M genes (Table 2). However, the diversity was larger since *bla*_CTX-M-1_ (*n* = 6), *bla*_CTX-M-15_ (*n* = 6), *bla*_CTX-M-14_ (*n* = 5) and *bla*_CTX-M-27_ (*n* = 1) were identified. The ESBL gene was mostly located on plasmids (IncF, IncI1), while it was identified on the chromosome in only one isolate. The IncF plasmids presented diverse formulas; on the contrary, all IncI1 plasmids carrying the *bla*_CTX-M-1_ gene belonged to the pST3 type, while the only IncI1 plasmid carrying the *bla*_CTX-M-15_ gene belonged to the pST37 type. Two out of the four *K. pneumoniae* carried carbapenemase genes in addition to the *bla*_CTX-M-15_ gene: #43716 displayed a *bla*_NDM-1_ gene on an IncF plasmid, while #43717 presented both *bla*_NDM-1_ and *bla*_OXA-48_ genes, respectively, located on an untypeable and on an IncL plasmid.

### 3.3. Resistance Genes to Non-Beta-Lactam Antimicrobial Agents

All isolates were considered MDR according to the definition by Magiorakos et al. since they presented resistance to at least two antibiotic families in addition to beta-lactams (Appendix A). All nine isolates from fish were also resistant to tetracyclines (*tetA* or *tetB* genes) and trimethoprim (genes belonging to the *dfr* family), whereas most of them were also resistant to aminoglycosides and sulfonamides. The most frequently identified family genes were *dfrA* and *aadA* conferring resistances to trimethoprim and aminoglycosides.

## 4. Discussion

Our study revealed that 1.4% of fish and 1.6% of clams bought from retail markets in Tunisia were contaminated with ESBL-producing Enterobacterales. Two previous Tunisian studies reported much higher proportions of ESBL-producing Enterobacterales in seafood: the first study identified ESBL-producing bacteria in 65% (52/80) of the tested pools of mussels, 8.3% (3/36) of oysters, and 14.4% (26/181) of clams [8], while the second one reported 46.8% of ESBL-producing Enterobacterales in fish from the Bizerte lagoon [19]. The proportions found in this study were lower than what was reported from fish in India (70% of *E. coli* and 25% of *K. pneumoniae*) [20] and from imported fish in Saudi Arabia (27.2%, 110/405) [21], similar to what was observed from farmed fish in China (1.5%, 3/218) [22], and higher than what was reported from seafood in Norway (2/549, 0.4%) [6] and from *Klebsiella* spp. recovered from freshwater fish in Hong Kong (1/476, 0.2%) [23]. However, comparison of proportions is difficult due to the diversity of protocols used, in terms of the origin of the samples (different species of fish, shellfish,…), methodology (with or without selection on antibiotic-containing media), and isolated bacteria (focus on specific species or large identification of all bacterial species present in the sample). To circumvent these limitations in comparison, it would be of high interest for both authorities and consumers to set up a monitoring program in this sector, which would provide resistance trends over the years.

Two out of the four *K. pneumoniae* identified here also displayed carbapenemase genes, which is a much scarcer feature than ESBL-producing Enterobacterales in seafood products. A KPC-3 ST167 *E. coli* was reported from mussels in Tunisia [24], and carbapenem-resistant isolates were sporadically identified in Germany (VIM-1 from a clam harvested in Italy) [25], in Myanmar (NDM-1 from one prawn and one clam sample, and IMI-1 from one fish) [26], or in Canada (one NDM-1- and three IMI-producing *Enterobacter cloacae* in shrimps and clams imported from Vietnam) [7].

The *bla*_CTX-M-1_ gene, which is usually related to an animal origin [27,28], was identified in six *E. coli* isolates belonging to different STs. Interestingly, four out of the six *bla*_CTX-M-1_ genes identified here were carried by an IncI1/pST3 plasmid, possibly suggesting transfers of this plasmid between different *E. coli* isolates. This plasmid, which has often been associated with the occurrence of the *bla*_CTX-M-1_ gene in animals including in Tunisia [28,29,30], was the only one recurrently identified in this study. The *bla*_CTX-M-1_-carrying isolates might come either from farms discharging effluents in the nearby rivers or from seabirds’ droppings. Even though we acknowledge that it is very difficult to trace the origin of a strain, such sources of contamination were hypothesized in other contexts [31,32].

On the other hand, the *bla*_CTX-M-15_ gene, which is commonly of human origin, was identified in all fish and one-third (6/18) of clam isolates, mostly carried by IncF plasmids. It was carried in only one case by an IncI1/pST37 plasmid, a pMLST type associated with the human host [33], contrary to the pST3 which is mainly found in animals. Likewise, several STs identified here were strongly associated with the human host. This is especially the case for *E. coli* ST131, which is a worldwide disseminated clone notably responsible for urinary infections [34]. In our study, two clams presented an *E. coli* ST131, displaying either the *bla*_CTX-M-27_ or *bla*_CTX-M-15_ ESBL genes. ST617 is also recurrently associated with human infections, notably blood and urinary tract infections, including in Tunisia [35,36]. Here, this ST was detected both in fish and clams. In fish, an identical ST617 clone was recovered from three different retail markets, which all had different suppliers. Nevertheless, the hypothesis of a common source is the most likely one.

The four *K. pneumoniae* belonged to multi-resistant clones circulating in humans: ST17 is known to carry ESBL determinants [37], while ST307 and ST147 are high-risk clones that emerged in the mid-1990s and became worldwide vectors of carbapenemases [38,39]. In Tunisia, ST147 has been recurrently reported in hospital settings since at least 10 years, mostly carrying the *bla*_NDM-1_ but also the *bla*_OXA-48_ gene [40,41], and ST307 was more recently reported as a cause of a *bla*_NDM-1_-producing outbreak [42].

Our results suggest that seafood can be a reservoir of multi-drug resistant bacteria, possibly of human origin. Our hypothesis is that sewage effluents, of human and animal origin, are discharged in rivers near the sea and that the bacterial load is high enough to contaminate fish and shellfish that are nearby. Such transmission of resistant bacteria to fish through contaminated wastewater has been proven notably in Tanzania and Vietnam [43,44]. In our study, fish from farms R1 and R3 were raised in the open sea, fish from farm R2 were raised in closed tanks filled with seawater, and clams were not cultivated but harvested in the open sea. We know that all four sampling sites were close to discharge points of treated wastewater from sewage treatment plants that, among others, are treating wastewater from large capacity hospitals. The hypothesis of a contamination through water is thus plausible and reinforced by a publication showing that sewage effluents of Tunisia are a potential source of carbapenemase genes, the most abundant ones being *bla*_OXA-48_ and *bla*_NDM-1_ [45]. Another study performed in Sfax, Tunisia, showed that antibiotic residues can be found in effluents, and that fluoroquinolones and macrolides are those that threaten the environment the most [46]. Fish and shellfish can thus be either directly contaminated by resistant bacteria, or by non-pathogenic Enterobacterales that acquired resistance genes. Moreover, multi-resistant bacteria might be concentrated in seafood, and especially in filter-feeding organisms such as mussels or clams [6].

In conclusion, our study reported relatively low proportions of ESBL-producing Enterobacterales in fish (1.4%) and clams (1.6%), thus suggesting a low risk for the consumer. Nevertheless, we raised the issue first of the environmental contamination by all effluents that can bring resistant bacteria to the sea, and second the risk of creating a reservoir of resistant bacteria in seafood products that are intended for human consumption. Consequently, measures should be taken to prevent bacterial contamination of rivers in general, and the occurrence of multi-drug resistant bacteria in seafood should be monitored in the same way as in livestock products.

## Figures and Tables

**Table 1 microorganisms-10-01364-t001:** Characteristics of isolates collected from fish.

Isolation Date	Isolate	Origin	Species	PG ^d^-ST	ESBL	ESBL Localization	Additional Resistances
31 August 2014	** 40557 ** ^a^	R1 ^b^	*E. coli*	A-8149	CTX-M-15	IncF/F-:A-:B53	*bla_TEM-1B_*, *tetA*, *dfrA14*, *qnrS1*, *mdfA*
	** 40558 **	R1	*K. pneumoniae*	983	CTX-M-15	NT ^c^ plasmid	*bla_OXA-1_*, *bla_TEM-1B_*, *sul2*, *tetA*, *dfrA14*, *aac(6′)-Ib-cr*, *oqxA*, *oqxB*, *qnrB1*, *aac(3)-IIa*, *aph(3”)-Ib*, *aph(6)-Id*, *catB3*
1 September 2014	40595	R1	*E. coli*	A-617	CTX-M-15	IncF/F31:A4:B1	*bla_OXA-1_*, *sul1*, *sul2*, *tetB*, *dfrA17*, *aac(6′)-Ib-cr*, *aadA5*, *aph(3”)-Ib*, *aph(6)-Id*, *aac(3)-IIa*, *catB3*, *mdfA*, *mphA*
16 October 2014	** 40560 **	R2	*K. pneumoniae*	983	CTX-M-15	NT plasmid	*bla_OXA-1_*, *bla_TEM-1B_*, *sul2*, *tetA*, *dfrA14*, *aac(6′)-Ib-cr*, *oqxA*, *oqxB*, *qnrB1*, *aac(3)-IIa*, *aph(3”)-Ib*, *aph(6)-Id*, *catB3*
2 November 2014	** 40561 **	R1	*E. coli*	A-10	CTX-M-15	Chromosome	*sul2*, *tetA*, *dfrA1*, *aadA1*, *aph(3”)-Ib*, *aph(6)-Id*, *mdfA*
4 November 2014	40563	R1	*E. coli*	A-10	CTX-M-15	Chromosome	*sul2*, *tetA*, *dfrA1*, *aadA1*, *aph(3”)-Ib*, *aph(6)-Id*, *mdfA*
14 November 2014	** 40564 **	R3	*E. coli*	A-617	CTX-M-15	IncF/F31:A4:B1	*bla_OXA-1_*, *sul1*, *sul2*, *tetB*, *dfrA17*, *aac(6′)-Ib-cr*, *aadA5*, *aph(3”)-Ib*, *aph(6)-Id*, *aac(3)-IIa*, *catB3*, *mdfA*, *mphA*
27 March 2015	40601	R2	*E. coli*	A-617	CTX-M-15	IncF/F31:A4:B1	*bla_OXA-1_*, *sul1*, *sul2*, *tetB*, *dfrA17*, *aac(6′)-Ib-cr*, *aadA5*, *aph(3”)-Ib*, *aph(6)-Id*, *aac(3)-IIa*, *catB3*, *mdfA*, *mphA*
15 May 2015	** 40598 **	R2	*K. pneumoniae*	13	CTX-M-15	NT plasmid	*bla_OXA-1_*, *bla_TEM-1B_*, *sul1*, *sul2*, *tetA*, *dfrA12*, *aadA2*, *aac(6′)-Ib-cr*, *qnrS1*, *aac(3)-IIa*, *aph(6)-Id*, *catB3*, *mphA*

^a^ Bold underlined isolates were whole-genome sequenced; 40564, 40601, and 40595 were clonal as determined by PFGE. Only 40564 was sequenced. Likewise, 40561 and 40563 were clonal so only 40561 was sequenced. ^b^ R1, R2, and R3 correspond to three different retail markets. ^c^ NT: non-typable (CTX-M with a plasmidic localization as determined by Southern blot, but on a band that did not match with a plasmid probe). ^d^ PG: phylogroup (only for *E. coli* isolates).

**Table 2 microorganisms-10-01364-t002:** Characteristics of isolates collected from clams.

Date of Isolation	No. of Clams (No. of Pool)	Isolate ^a^	PG ^b^-ST	ESBL/	ESBL Localization	Additional Resistances
				Carbapenemase		
10 March 2016	220 (44)	Ec-**43697**	B2-131	CTX-M-27	IncF/F1:A2:B20	*sul1*, *sul2*, *tetA*, *dfrA17*, *aadA5*, *aph(3”)-Ib*, *aph(6)-Id*, *mdfA*, *mphA*
		Ec-**43698**	D-38	CTX-M-14	IncF/F1:A-:B23	*dfrA1*, *aadA1*, *mdfA*
		Kp-**43699**	307	CTX-M-15	IncF/F7°:A-:B-	*bla_OXA-1_*, *sul2*, *tetA*, *dfrA14*, *aac(6′)-Ib-cr*, *oqxA*, *oqxB*, *qnrB1*, *aph(3”)-Ib*, *aph(6)-Id*, *aac(3)-IIa*, *catB3*,
22 March 2016	160 (32)	Ec-**43701**	A-617	CTX-M-14	IncI2	*bla_TEM-1B_*, *sul1*, *sul2*, *dfrA17*, *dfrA8*, *aadA1*, *aad15*, *ant(2”)-Ia*, *aph(3”)-Ib*, *aph(6)-Id*, *floR*, *mdfA*, *mphA*
		Ec-43702	D-38	CTX-M-14	IncF/F1:A-:B23	*dfrA1*, *aadA1*, *mdfA*
		Ec-**43703**	D-602	CTX-M-1	IncI1/ST3	*sul2*, *tetB*, *dfrA17*, *aadA5*, *aph(3”)-Ib*, *mdfA*
		Ec-**43704**	B1-1196	CTX-M-1	IncI2	*bla_TEM-1B_*, *mdfA*
		Ec-**43707**	A-48	CTX-M-1	IncI2	*bla_TEM-1B_*, *sul3*, *tetA*, *dfrA1*, *aadA1*, *mdfA*
28 March 2016	270 (54)	Ec-**43710**	A-2253	CTX-M-1	IncI1/ST3	*sul2*, *tetA*, *dfrA14*, *mdfA*
		Ec-**43711**	A-8059	CTX-M-1	IncI1/ST3	*sul1*, *sul2*, *tetA*, *dfrA12*, *dfrA17*, *aadA1*, *aadA15*, *mdfA*, *mphA*
		Ec-**43712**	A-9512	CTX-M-15	IncI1/ST37	*bla_TEM-1B_*, *tetA*, *aac(3)-IId*, *mdfA*
		Ec-**43713**	B2-131	CTX-M-15	IncF/ F31:A4:B1	*bla_OXA-1_*, *sul1*, *dfrA17*, *aac(6′)-Ib-cr*, *aac(3)-IIa*, *aadA5*, *catB3*, *mdfA*, *mphA*
		Ec-**43714**	A-617	CTX-M-14	Chromosome	*bla_OXA-10_*, *bla_TEM-1B_*, *sul1*, *sul2*, *sul3*, *tetA*, *tetB*, *aadA1*, *ant(2”)-Ia*, *aph(3′)-Ia*, *mdfA*
		Ec-43715	D-38	CTX-M-14	IncF/F1:A-:B23	*dfrA1*, *aadA1*, *mdfA*
		Kp-**43716**	17	CTX-M-15/ NDM-1	IncHI1B/IncF	*bla_OXA-1_*, *bla_TEM-1B_*, *sul1*, *dfrA5*, *aac(6′)-Ib-cr*, *oqxA*, *oqxB*, *qnrB1*, *aac(3)-IIa*, *catA1*, *catB3*, *ereA*, *ereB*
4 April 2016	120 (24)	Kp-**43718**	147	CTX-M-15/ OXA-48, NDM-1	IncR/IncL, IncF	*bla_TEM-1B_*_,_*tetA*, *sul1*, *dfrA1*, *oqxA*, *oqxB*, *qnrB1*, *aph(3”)-Ib*, *aph(6)-Id*
		Kp-**43719**	147	CTX-M-15	IncF/F2:A22:B-	*bla_OXA-1_*, *bla_TEM-1B_*, *sul1*, *dfrA5*, *aac(6′)-Ib-cr*, *oqxA*, *oqxB*, *qnrB1*, *aac(3)-IIa*, *catA1*, *catB3*, *ereA*, *ereB*
22 April 2016	305 (61)	Ec-**43720**	A-2253	CTX-M-1	IncI1/ST3	*sul2*, *tetA*, *dfrA14*, *mdfA*

^a^ Ec: *E. coli*; Kp: *K. pneumoniae*. Bold underlined isolates were WG sequenced; 43698, 43702, and 43715 were clonal as determined by PFGE. Only 43698 was sequenced. ^b^ PG: phylogroup (only for *E. coli* isolates).

## Data Availability

The data presented in this study are available on request from the corresponding author.

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
