# Peer review of "Prevalence and Characterization of Extended-Spectrum β-Lactamase- and Carbapenemase-Producing Enterobacterales from Tunisian Seafood"

_microorganisms, 2022, doi:10.3390/microorganisms10071364_

Round 1
Reviewer 1 Report
Multi drug resistance is a major issue globally and focused surveillance studies are always helpful. This study focuses of the prevalence and characterization of ESBL and carbapenems producing (CPO) Enterobacterales isolated during the period of 2014-2016.
Comments:
1. As the authors are discussing about the occurrence of ESBL and CPO in seafoods ,its advisable to look into prevalence trend of the concerned organisms in present scenario as every year there is a huge change in the trend.
2.Line 24 , 55-56 , 63, 65 needs minor corrections .
Author Response
Comments:
- As the authors are discussing about the occurrence of ESBL and CPO in seafoods ,its advisable to look into prevalence trend of the concerned organisms in present scenario as every year there is a huge change in the trend.
Answer: we agree with the reviewer that having trends in resistance is this sector would be highly valuable. Trends cannot be inferred from our data since the study was not designed in that purpose. Our study was a first baseline, but monitoring resistance pheno/genotypes over years can surely be advised. This has been included in the revised version.
2.Line 24 , 55-56 , 63, 65 needs minor corrections .
Answer: it was not fully clear which corrections were needed. We did our best to answer to this question.
Reviewer 2 Report
Sola et al characterised ESBL and carbapenemase producing Enterobacterales by whole genome sequencing and antibiotic resistance profiling in seafood samples from Tunisia. A low prevalence (< 2%) of ESBL bacteria was found in farmed fish and clams associated with the carriage of blaCTX-M genes in E. coli and K. pneumoniae. The blaCTX-M-15 and blaCTX-M-15 genes were typically associated with ESBL carriage located on plasmids. Two isolates sampled from clams carried known carbapenemase genes (blaNDM-1 25 and blaOXA-48). Several STs of MDR genotypes implicated in human infection were found. The authors suggest that farm and hospital effluent may be responsible for contamination of seafood by MDR bacteria.
The publication would benefit from moderate revision to improve the content. References to points made are lacking and more details of the methods used provided. The regional origins of the clams are not provided (market sources were listed in the fish) so it is difficult to assess the distribution of the clones (could this be provided in a Figure with colour coded MDR phenotype?). From the WGS are common mobile elements present in the MDR phenotype which would increase risk of transmission? One drawback of the study, is that the authors conclude that environmental effluent (farm and hospital, and human sewage also?) is contributing to the contamination of farmed seafood, however, no such analysis of effluent was included in the present study. This speculation requires further elaboration in the Discussion and/or further interrogation of the genome structure and comparison to previous studies would help to assess the MDR risks and clonal origins. The present text requires a more balanced point of view addressing the significance of the findings and likely explanation of MDR origin.
Please find my suggested edits below to improve the text
Line 5 Sola or Soula used in citation
Line 22 Escherichia in full
Line 24 animal
Line 28 replace organisms, farmed species?
Line 33 Introduction lacks References throughout
Line 42 contrary
Line 48 coastline singular
Line 55 Improve sense, spelling and style of sentence
Line 68 Line 128 fish singular
Line 70, Line 76 state amount of tissue and broth volume used, any homogenisation?
Line 73 provide breakdown of location and origin of clam markets similar to fish
Line 79 2 mg/ml
Line 80 how many colony types on average?
Line 85 provide disc concentration in Suppl Table?
Line 88 quality control strain
Line 90 reference DDST, state ertapenem disc concentration
Line 98 reference interpretation criteria
Line 116 reference and check sense, adequate probes?
Line 117 check style
Line 127 replace 'ones 'with specific term eg isolates
Line 129 reduce sentence using (n=)
Line 132 provide reference phylogroup A non-pathogenic status
Table 1 & 2 Additional resistance gene classes as subheadings to make distribution clearer in columns?
Line 137 explain non-typeable outcome
Line 144 isolate singular
Line 157 untypeable
Line 163 replace plasmidic
Line 165 provide Suppl Table to list phenotypic resistance found to ESBL, carbapenem and non-beta lactams and methods used
Line 167 resistance singular
Line 174 In previous studies
Line 178 use specific term eg study, replace 'one'
Line 179 state or group studies that used selective media and similar target organisms otherwise comparison of prevalence is difficult
Line 198 provide reference of animal origin
Line 199 evidence and reference origin statement
Line 205 provide reference, delete space
Line 208 infection types?
Line 209 insert space
Line 211 provide evidence for common source and reference origin of other ST617
Line 219 present case for your statement and reference effluent studies, seafood studies, antibiotic residue analysis to expand hypothesis
Line 220 human effluent is also contributory?
Line 224 check sense rephrase
Line 234 re-phrase as this study does not look for environmental isolates (effluent) or MDR genes contained within, discuss 'potential' risk of environmental transmission and reference relevant studies assessing distribution of MDR in different environmental compartments (animal, man, water, soil), and control measures eg need to reduce antibiotic use also?
Line 239 delete full stop
Line 229, Line 230 provide references throughout paragraph for each point made
Author Response
The publication would benefit from moderate revision to improve the content. References to points made are lacking and more details of the methods used provided.
Answer: these points were addressed below in the answers to the specific points.
The regional origins of the clams are not provided (market sources were listed in the fish) so it is difficult to assess the distribution of the clones (could this be provided in a Figure with colour coded MDR phenotype?).
Answer: we understand the suggestion. However, since there are only 4 sampling cities, we fear that such a map would not be informative. We propose to only keep the tables, unless the editor thinks otherwise.
From the WGS are common mobile elements present in the MDR phenotype which would increase risk of transmission?
Answer: we agree with the reviewer that this has not been discussed. IncF plasmids seem to be quite diverse. On the contrary, all blaCTX-M-1 genes were carried by IncI1 ST3 plasmids. Consequently, this plasmid might play a particular role in the transmission of this gene. This has been included in the discussion and results were also completed. We than the reviewer for this interesting remark.
One drawback of the study, is that the authors conclude that environmental effluent (farm and hospital, and human sewage also?) is contributing to the contamination of farmed seafood, however, no such analysis of effluent was included in the present study. This speculation requires further elaboration in the Discussion and/or further interrogation of the genome structure and comparison to previous studies would help to assess the MDR risks and clonal origins.
Answer: as wisely requested by the reviewers, we introduced references to strengthen our hypotheses and tuned down our arguments which, as was recurrently repeated, are only hypotheses.
The present text requires a more balanced point of view addressing the significance of the findings and likely explanation of MDR origin.
Answer: we paid attention to tune down our arguments throughout the text.
Please find my suggested edits below to improve the text
Line 5 Sola or Soula used in citation
Answer: the correct name is Sola, this was corrected.
Line 22 Escherichia in full
Answer: this was expanded as suggested.
Line 24 animal
Answer: this was corrected.
Line 28 replace organisms, farmed species?
Answer: this was changed for “seafood and fish” to avoid any confusion with the livestock farms mentioned in the same sentence.
Line 33 Introduction lacks References throughout
Answer: references were added to strengthen our introduction
Line 42 contrary
Answer: this was corrected.
Line 48 coastline singular
Answer: this was corrected.
Line 55 Improve sense, spelling and style of sentence
Answer: the sentience was modified. We hope this is now satisfying.
Line 68 Line 128 fish singular
Answer: this was modified.
Line 70, Line 76 state amount of tissue and broth volume used, any homogenisation?
Answer: the amount of tissue cannot be stated exactly. The whole intestine was used for each fish. It was placed in 10 ml of peptone broth and homogenized before incubation. This has been introduced in the revised M&M section.
Line 73 provide breakdown of location and origin of clam markets similar to fish
Answer: clams originated from Gabès, in the south east of Tunisia.
Line 79 2 mg/ml
Answer: this was indeed mg/ml or mg/L.
Line 80 how many colony types on average?
Answer: one morphology per plate was recovered on most of the plates. However, a few of them presented 2 colony morphologies, which were both picked up for further characterization.
Line 85 provide disc concentration in Suppl Table?
Answer: a supplemental Table S1 with all tested antibiotics and disc concentration was included as requested.
Line 88 quality control strain
Answer: this was modified.
Line 90 reference DDST, state ertapenem disc concentration
Answer: a reference was added for the DDST test. An ertapenem disc of 10 mg was used. This was included in the revised version.
Line 98 reference interpretation criteria
Answer: the website was replaced by a reference.
Line 116 reference and check sense, adequate probes?
Answer: a reference was added. The term “adequate” encompassed both genes and plasmids of interest for this study. This was included in the revised version.
Line 117 check style
Answer: the sentence was modified. We hope this is now clearer.
Line 127 replace 'ones 'with specific term eg isolates
Answer: this was modified.
Line 129 reduce sentence using (n=)
Answer: this was modified.
Line 132 provide reference phylogroup A non-pathogenic status
Answer: the reference by Clermont et al (2000) was added.
Table 1 & 2 Additional resistance gene classes as subheadings to make distribution clearer in columns?
Answer: we propose to leave the tables as they are. Otherwise, considering the high number of resistance genes in certain isolates, assigning one column to each resistance gene would make the table far too large to fit the journal format. Nevertheless, since the reviewer asked for a table S3 compiling phenotypic results, genes were included in column in this new table.
Line 137 explain non-typeable outcome
Answer: a plasmid is considered as non-typeable if one band of the PFGE-S1 gel gave a positive signal with a CTX-M probe but not with a plasmid probe. This was included in the table legend.
Line 144 isolate singular
Answer: this was modified.
Line 157 untypeable
Answer: this was modified.
Line 163 replace plasmidic
Answer: the sentence was modified.
Line 165 provide Suppl Table to list phenotypic resistance found to ESBL, carbapenem and non-beta lactams and methods used
Answer: a new Table S3 was provided as requested, which compiles phenotypic and genotypic characteristics of all isolates.
Line 167 resistance singular
Answer: this was modified.
Line 174 In previous studies
Answer: the sentence was modified.
Line 178 use specific term eg study, replace 'one'
Answer: we replaced the first “one” by “studies’” to be clear. But we proposed to leave “the second one” since it clearly refers to the second study.
Line 179 state or group studies that used selective media and similar target organisms otherwise comparison of prevalence is difficult
Answer: we agree with the reviewer that comparison is difficult. But grouping studies is also impossible since no study used the same design or methodology. We thus propose to leave the text as such, with the limitations clearly highlighted;
Line 198 provide reference of animal origin
Answer: two references were added as requested.
Line 199 evidence and reference origin statement
Answer: references were included to strengthen our hypothesis.
Line 205 provide reference, delete space
Answer: the reference was wrongly added after the next sentence. This reference was moved at its correct place.
Line 208 infection types?
Answer: the references reported (when stated) blood and urinary tract infections. This has been introduced in the revised version.
Line 209 insert space
Answer: this was modified.
Line 211 provide evidence for common source and reference origin of other ST617
Answer: we cannot bring more evidence here. Nevertheless, detecting three times the same clone in three markets was surprising. Hence, we hypothesized that these markets had a common intermediate. But this is only a hypothesis.
Line 219 present case for your statement and reference effluent studies, seafood studies, antibiotic residue analysis to expand hypothesis
Answer: references were added to strengthen our hypothesis.
Line 220 human effluent is also contributory?
Answer: yes, human effluents can also have a contribution here. However, we have no idea about the different contributions. The sentence was thus modified to put human and animal sources on the same level.
Line 224 check sense rephrase
Answer: the sentence was modified and we hope this is now clear
Line 234 re-phrase as this study does not look for environmental isolates (effluent) or MDR genes contained within, discuss 'potential' risk of environmental transmission and reference relevant studies assessing distribution of MDR in different environmental compartments (animal, man, water, soil), and control measures eg need to reduce antibiotic use also?
Answer: we propose to let this sentence without further references since it is meant to conclude the study. References were included in the previous paragraphs.
Line 239 delete full stop
Answer: this was modified.
Line 229, Line 230 provide references throughout paragraph for each point made
Answer: a reference was provided.

Reviewer 3 Report
Dears authors
Aquaculture is a growing sector where it is important to monitor for the presence of multi-resistant bacteria (MDR). The presence of broad spectrum b-lactamases (ESBL-) or carbapenemase-producing Enterobacterales is a commonly used indicator of resistance load in a given sector. In this study, 641 pieces of farmed fish (sea bream and sea bass) and 1075 Mediterranean clams were analyzed. All collected ESBL- and carbapenemase-producing Enterobacteria were sequenced throughout the genome. The proportion of ESBL-producing Enterobacterales was 1.4% in fish and 1.6% in clams, transported by E. coli (n = 23) and Klebsiella pneumoniae (n = 4). This phenotype was due solely to the presence of the blaCTX-M genes, the most frequent of which was blaCTX-M-15. The blaCTX-M-1 gene has also been identified in six E. coli, probably betraying an animl origin. Carbapenemases were absent in fish but identified in two isolates of K. pneumoniae from clams (blaNDM-1 and blaOXA-48). Several identified sequence types (STs) have been associated with human MDR clones such as E. coli ST131 and ST617 or K. pneumoniae ST307 and ST147. Monitoring of the quality of water discharged into the sea and the presence of MDR bacteria in seafood is mandatory to ensure the quality of fishery products.
- deepen the phenotypic evaluations with tables, in particular the subtitl 3.1 and table 1, using the following papers as an example and then reporting them in the bibliography : PMID: 34572716 ; PMID: 35321081
- Revision language e syntax's for manuscript
Author Response
- deepen the phenotypic evaluations with tables, in particular the subtitl 3.1 and table 1, using the following papers as an example and then reporting them in the bibliography : PMID: 34572716 ; PMID: 35321081
Answer: as also suggested by reviewer 1, a new Table S3 was provided, which compiles phenotypic and genotypic characteristics of all isolates. We propose not to introduce the references since they only concern Pseudomonas isolates, a bacterial species which is not discussed here.
- Revision language e syntax's for manuscript
Answer: we made our best to revise the English in the short period of time given for the revision.